# Alveolar proteins stabilize cortical microtubules in *Toxoplasma gondii*

Clare R. Harding[1], Matthew Gow[2], Joon Ho Kang[3,4], Emily Shortt[1], Scott R. Manalis [4,5,6], Markus Meissner[2,7] & Sebastian Lourido [1,8]

Single-celled protists use elaborate cytoskeletal structures, including arrays of microtubules at the cell periphery, to maintain polarity and rigidity. The obligate intracellular parasite *Toxoplasma gondii* has unusually stable cortical microtubules beneath the alveoli, a network of flattened membrane vesicles that subtends the plasmalemma. However, anchoring of microtubules along alveolar membranes is not understood. Here, we show that GAPM1a, an integral membrane protein of the alveoli, plays a role in maintaining microtubule stability. Degradation of GAPM1a causes cortical microtubule disorganisation and subsequent depolymerisation. These changes in the cytoskeleton lead to parasites becoming shorter and rounder, which is accompanied by a decrease in cellular volume. Extended GAPM1a depletion leads to severe defects in division, reminiscent of the effect of disrupting other alveolar proteins. We suggest that GAPM proteins link the cortical microtubules to the alveoli and are required to maintain the shape and rigidity of apicomplexan zoites.

[1] Whitehead Institute for Biomedical Research, Cambridge 02142 MA, USA. [2] Wellcome Centre for Molecular Parasitology, Institute of Infection, Immunity & Inflammation, University of Glasgow, Glasgow G12 8TA, UK. [3] Department of Physics, Massachusetts Institute of Technology, Cambridge 02139 MA, USA. [4] Koch Institute for Integrative Cancer Research, Massachusetts Institute of Technology, Cambridge 02139 MA, USA. [5] Department of Biological Engineering, Massachusetts Institute of Technology, Cambridge 02139 MA, USA. [6] Department of Mechanical Engineering, Massachusetts Institute of Technology, Cambridge 02139 MA, USA. [7] Department of Veterinary Sciences, Ludwig-Maximilians-Universität, Munich 80539, Germany. [8] Biology Department, Massachusetts Institute of Technology, Cambridge 02139 MA, USA. Correspondence and requests for materials should be addressed to C.R.H. (email: harding@wi.mit.edu) or to S.L. (email: lourido@wi.mit.edu)

Microtubules lend structure, rigidity, and polarity to processes ranging from cell division to vesicular trafficking in eukaryotic cells. Unicellular organisms often display elaborate cytoskeletal adaptations that frequently involve the organisation of membranes around microtubule arrays[1–4]. Cortical microtubule arrays are frequently found subtending the plasmalemma of diverse organisms including alveolates, trypanosomes, and flatworm spermatozoa. These structures are assumed to provide the stability and rigidity these organisms need to withstand the forces they experience over their complex life cycles.

Cortical microtubule arrays are present in most motile stages of alveolates belonging to the phylum Apicomplexa[5], which includes important pathogens such as *Plasmodium* spp., the causative agents of malaria, and *T. gondii*, a ubiquitous parasite of warm-blooded animals and an opportunistic pathogen of humans. In addition to cortical microtubules (also known as subpellicular microtubules), the asexual stages of *T. gondii* harbour four other distinct tubulin-containing structures: centrioles, a conoid, intraconoid microtubules, and a spindle in replicating parasites[6]. *T. gondii* has 22 cortical microtubules that extend over two thirds of the length of the cell body, originating from the apical polar ring, which acts as an atypical microtubule organizing centre[7]. Unlike mammalian microtubules, cortical microtubules from *T. gondii* are exceptionally stable, resisting depolymerisation upon detergent extraction and cold treatment[8,9]. These unusual properties are associated with parasite-specific microtubule-associated proteins (MAPs) that decorate the microtubules[10], including SPM1/2, TrxL1/2, and TLAP1/2/3/4[11-13]. MAPs display a surprising level of functional redundancy. Cortical microtubules remain largely normal upon knockout of various MAPs—even upon double or triple knockouts—suggesting that the MAPs tested are not responsible for microtubule arrangement or polymerisation, or that their functions are highly redundant. Nonetheless, deletion of SPM1 or triple deletion of SPM1, TLAP2, and TLAP3 increased microtubule sensitivity to detergent extraction or cold treatment, respectively[11,12].

It is not known how cortical microtubules remain anchored and arranged along membranes. Apicomplexan cortical microtubules are tightly attached to the alveoli. The sub-pellicular network (SPN) lies beneath the alveoli and is formed from a stable network of intermediate filament-like proteins. The alveoli, SPN, and supporting microtubules are termed the inner membrane complex (IMC). Ultrastructural studies have documented the tight connections between the constituents of the IMC. Inner membranous particles (IMPs) are arranged along the alveoli following the path of the microtubules with a 32 nm periodicity that matches that of the MAPs that decorate cortical microtubules[8,10]. Cryo-electron tomography has revealed apparent protein linkers that bridge the 27 nm space between the microtubules and the alveolar membrane[14]. Both IMPs and the linkers extend beyond the length of the microtubules[8,10], suggesting that the structures for ordering and attaching microtubules originate from the alveoli; however, no alveolar membrane proteins have been specifically localised to such periodic structures or implicated in microtubule attachment.

The IMC plays a key role in *T. gondii* replication, which occurs through endodyogeny, whereby the cytoskeletons and organelles for two daughter cells are formed in the maternal cytoplasm before cytokinesis[15]. Genetic perturbation of structural alveolar proteins[16,17] or components of the SPN[18], or chemical disruption of microtubules[19,20] all prevent proper daughter cell elongation and budding. Furthermore, disruption of IMC components can alter daughter cell orientation[21] or number[17,18]. In addition, deletion of PhIL1, an integral IMC protein, results in shorter cells[22], and certain tubulin mutations can result in longer

parasites[23], suggesting that the IMC, and cortical microtubules in particular, are major determinants of parasite shape.

The IMC appears to additionally support vesicular trafficking and motility. Secretory vesicles called micronemes align along cortical microtubules[7], and the *T. gondii* mitochondrion makes extensive contacts with the IMC[24]. Moreover, the acto-myosin motor complex (glideosome) that drives motility is anchored to the IMC[25,26]. The precise role of these scaffolding functions has been difficult to ascertain, as deletion of IMC-resident proteins either results in no phenotype[17,18,27] or causes gross structural alterations that obscure specific cellular processes[16,17]. Furthermore, conditional disruption of IMC proteins has thus far employed systems that require multiple rounds of parasite replication to take effect, complicating the interpretation of the phenotypes observed[16,27,28].

Deletion of glideosome components predicted to anchor the complex to the IMC—like the peripherally associated GAP45[28] or the integral proteins GAP40 and GAP50[16]—caused severe defects in the morphology of replicating parasites, reminiscent of other structural defects in the IMC[17]. Such defects are not observed upon deletion of glideosome components that directly participate in motility, such as MyoA, MLC1, or ELC1[28,29]. We therefore sought to characterize the role of the GAPM (glideosome-associated protein with multiple-membrane spans) family, which co-immunoprecipitate with glideosome and SPN components, have been localized to the IMC[30], and appear to affect parasite morphology[16]. We used the auxin-induced degron (AID) system to conditionally deplete members of the GAPM family with rapid kinetics. We show that both GAPM1a and GAPM2a are independently essential for the parasite's life cycle, leading to rapid changes in vacuole and parasite morphology and profound cytoskeletal defects. These observations implicate GAPM proteins as major components of the apicomplexan cytoskeleton and essential for maintaining the structural stability of parasites.

## Results

**GAPM proteins localise to the IMC.** The GAPMs comprise a family of conserved apicomplexan proteins distributed in three main clades and characterized by five or six transmembrane domains (Fig. 1a). Previous studies localised several family members to the IMC of *T. gondii* and *Plasmodium falciparum* by overexpressing fusion proteins[30]. To further investigate these proteins, we C-terminally tagged the endogenous loci with YFP or mCherry and monitored their localisation using super-resolution structured illumination microscopy (SR-SIM). All GAPMs were observed at the IMC in mature cells as well as the developing daughter cells (Fig. 1b). GAPM1a, GAPM2a, and GAPM3 were highly expressed, while GAPM1b and GAPM2b could only be observed using antibodies against the fluorophores. This expression pattern matches their transcriptional profiles (ToxoDB; Fig. 1c). Furthermore, CRISPR-based genome-wide screens revealed that GAPM1a, GAPM2a, and GAPM3 contribute to parasite fitness during growth in fibroblasts, while GAPM1b and GAPM2b are dispensable[31] (Fig. 1c). Based on their low expression and predicted dispensability, GAPM1b and GAPM2b were not investigated further.

We observed accumulation of GAPM1a-YFP, GAPM2a-mCherry, and GAPM3-YFP at ring-like structures in the maternal IMC (Supplementary Figure 1a). These structures were ~250 nm wide and could not be resolved using conventional widefield microscopy. Co-expression of endogenously tagged GAPM3-YFP with either GAPM1a-mCherry or GAPM2a-mCherry demonstrated that the proteins colocalised at the IMC and were similarly enriched within the ring structures (Fig. 1d). Dually tagged strains could not be maintained in culture,

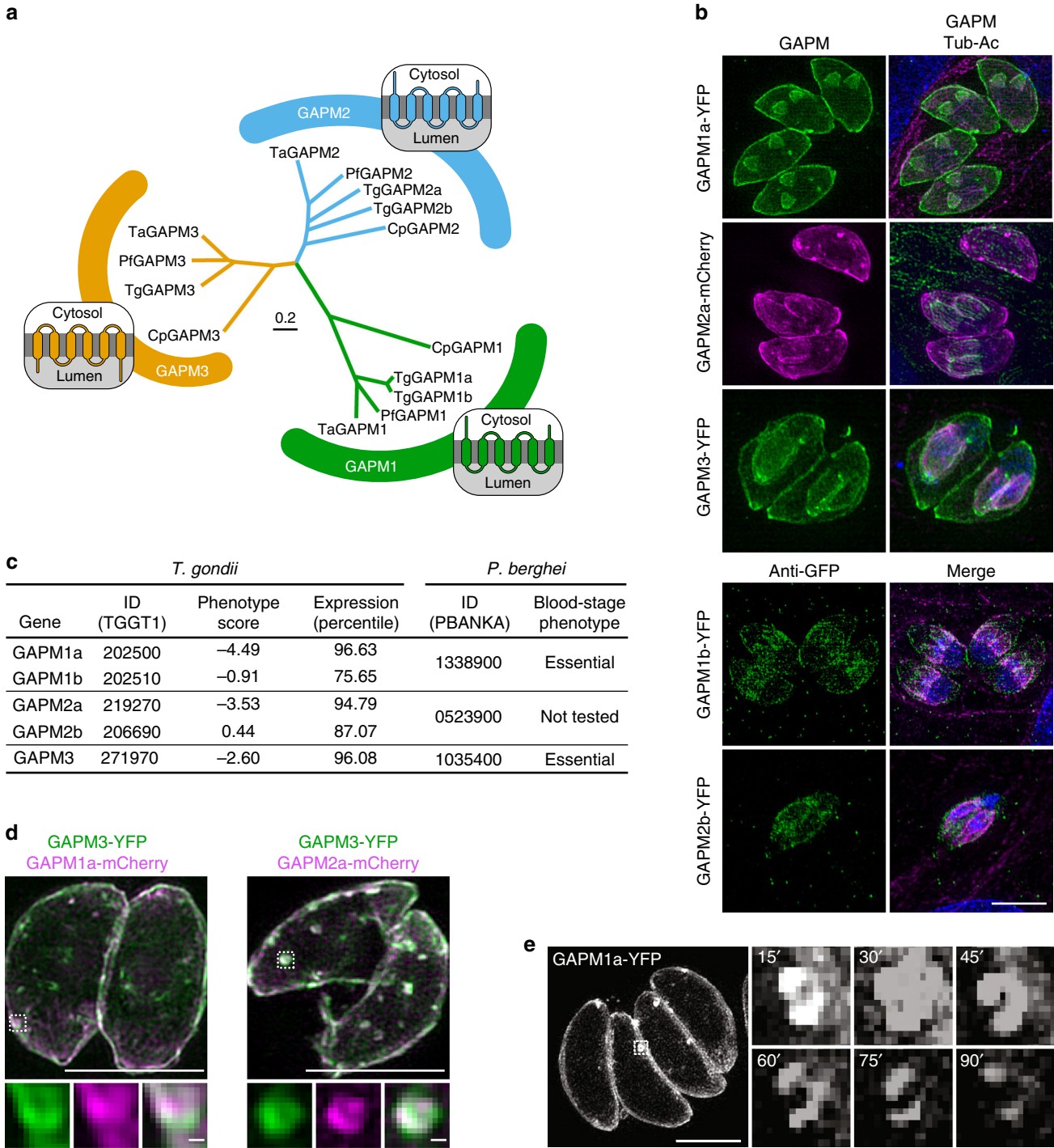

**Fig. 1** GAPM proteins localise to the IMC and are enriched at ring-like structures. **a** Cladogram of the GAPM proteins conserved among apicomplexans. Predicted membrane topology for each clade is illustrated[30]. **b** Table of GAPM genes, their transcriptional expression, and their importance for tachyzoite-stage *T. gondii*[31] and blood-stage *P. berghei*[32]. **c** GAPM proteins were C-terminally tagged with YFP or mCherry and visualised by SR-SIM in relation to acetylated tubulin. Projections of z-stacks of dividing cells are shown. GAPM1b-YFP and GAPM2b-YFP were expressed at a low level and therefore visualised indirectly with antibodies against GFP. All proteins were localised to the IMC in both mature and dividing cells. **d** Endogenously tagged GAPM proteins co-localise throughout the IMC and at ring structures (detail). Detail scale bar is 100 nm. **e** Live-cell SR-SIM of GAPM1a-YFP. Z-stacks were acquired every 7 min and projected to show fixed ring structures within the IMC that do not change in size or position over the course of 90 min. Images are representative of three independent experiments. Detail scale bar is 500 nm. Unspecified scale bars are 5 μm

although individual tags were well tolerated, suggesting a cumulative effect of tagging the various GAPM proteins. We examined the behaviour of the rings in live parasites expressing GAPM3-YFP, revealing that the structures remained immobile in the IMC for up to 90 min—the maximum period of observation

before the fluorophores were bleached (Fig. 1e, Supplementary Movie 1). Deformations of the plasma membrane, stained with the lipophilic dye PKH26, coincided with the site of GAPM rings, and the plasma membrane could occasionally be seen traversing the structure into the lumen of the parasite (Supplementary

Figure 1b). These rings were most prevalent at suture sites between alveolar plates, marked by endogenously HA-tagged ISC3[33] (Supplementary Figure 1c and d). While we cannot exclude the possibility that these structures are artefacts of protein tagging, they may represent openings in the IMC—perhaps the micropores previously visualised by electron microscopy[34]—where concentration of the alveolar membranes leads to signal amplification for integral membrane proteins. Apart from these foci, GAPMs are evenly distributed throughout the IMC.

**GAPM1a is essential for the completion of the lytic cycle.** Defects in replication can obscure the direct consequences of disrupting IMC proteins[16,33]. Systems for the rapid, conditional depletion of proteins require that the protein of interest be recognized by degradation machinery in the cytosol. To empirically determine the localisation of GAPM C termini in live parasites, we expressed a cytosolic single-chain nanobody that recognises GFP and YFP[35], fused to mCherry for visualisation and a destabilization domain for regulated expression[36] (Fig. 2a). As expected, the nanobody was recruited by fusion proteins where GFP or YFP was accessible to the cytosol, such as histone 2B (H2B-GFP)[37] and a mitochondrial surface marker (OMP-GFP, Fig. 2b). In contrast, the nanobody remained diffuse throughout the cytosol in the absence of a tagged protein or when GFP was sequestered in the mitochondrial matrix (SOD2-GFP, Fig. 2b). We observed robust recruitment of the nanobody to the IMC of parasites expressing either GAPM1a-YFP or GAPM3-YFP (Fig. 2b), establishing that the C termini of both proteins are cytosolic.

The topology of GAPM1a therefore enables us to conditionally regulate its expression using the AID system recently adapted for *T. gondii*[38]. This system adopts a plant-specific degradation pathway regulated by the phytohormone indole-3-acetic acid (IAA). When IAA is added to the media, the constitutively expressed *Oryza sativa* F-box ubiquitin ligase Tir1 targets proteins tagged with a small auxin-inducible degron (AID). This results in the regulated ubiquitination and rapid degradation of the AID-tagged protein (Fig. 2c). Endogenous GAPM1a was C-terminally tagged with mNeonGreen fused to a minimal AID and Ty epitope to generate the GAPM1a-AID strain. Immunoblotting against the Ty epitope confirmed that GAPM1a protein levels were reduced as soon as 2 h after the addition of IAA, and largely undetectable after 18 h (Fig. 2d).

To determine the effect of prolonged GAPM1a degradation, we performed plaque assays in the presence or absence of IAA. Due to a slight growth delay in the GAPM1a-AID strain, its plaques were best observed nine days post infection (Fig. 2e). As expected, addition of IAA did not affect the parental Tir1 strain. However, GAPM1a-AID parasites formed no plaques in the presence of IAA, indicating that GAPM1a is essential for parasite viability. In order to examine the phenotype more closely, GAPM1a-AID parasites were treated with IAA for different lengths of time prior to fixation. Depletion of GAPM1a was not uniform along the longitudinal axis of the parasite (Fig. 2f). GAPM1a signal was weaker in the middle section of the IMC after 1 h of IAA treatment and fully depleted from that region after 2–4 h of treatment. However, signal persisted at the apical and basal ends of the parasites during depletion, possibly due to inaccessibility of these highly-structured compartments to the proteasome or an altered conformation of GAPM1a, which partially occludes the AID tag. The complete depletion of GAPM1a from the body of the parasites coincided with the emergence of abnormal vacuoles 4–6 h after the addition of IAA. After 6 h of IAA treatment, 75.5 % (95% CI 68.64–82.36) of vacuoles exhibited IMC abnormalities or had severe morphological defects (Fig. 2g). After 18 h of IAA

treatment, vacuoles contained unstructured, multi-nucleated cells that exhibited dissociation of GAP45 and IMC1 at multiple places (arrowheads, Fig. 2f, g). A similar phenotype of unstructured, multi-nucleate cells was previously observed to coincide with parasite replication after deletion of GAP40 or GAP50[16,17]. The long-term effects of GAPM1a depletion are therefore consistent with other perturbations that alter the structure or biogenesis of the IMC.

To further examine the IMC ultrastructure during GAPM1a depletion, we performed transmission electron microscopy (TEM). At 6 h IAA treatment, many parasites appeared largely normal with the double membrane of the alveoli clearly distinguishable (Fig. 2h). However, after 18 h of IAA treatment, GAPM1a-AID parasites contained multiple nuclei and apical structures. Prolonged depletion of GAPM1a caused vesiculation of the alveoli, leaving large gaps in the IMC (detail, Fig. 2h).

We also explored the predicted essentiality of GAPM2a by tagging it with mNeonGreen-AID. GAPM2a-AID-tagged parasites had a more severe growth defect than GAPM1a-AID and formed very small plaques. Despite the growth defect, clones were isolated in which we could inducibly deplete GAPM2a (Supplementary Figure 2a), blocking plaque formation (Supplementary Figure 2b). Depletion of GAPM2a-AID followed similar kinetics to GAPM1a, and the protein was undetectable outside of the apical and basal ends 4 h after IAA addition. The morphological changes observed following GAPM2a depletion resembled those described for GAPM1a (Supplementary Figure 2a). We therefore conclude that GAPM2a is independently essential for the parasite's lifecycle.

**GAPM1a loss does not significantly affect parasite motility.** We took advantage of the rapid depletion of GAPM1a-AID to test its function 2–4 h after the addition of IAA, when GAPM1a-AID is undetectable throughout most of the IMC but replication defects are still rare (Fig. 2f). Based on the prior association of GAPM proteins with components of the glideosome, we investigated whether acute GAPM1a depletion affected motility or invasion. Parasites were treated with IAA for 2 or 4 h while intracellular, mechanically released, and used in trail deposition assays to measure 2D motility. Depletion of GAPM1a did not alter the frequency of trail formation (Fig. 3a), although subtle qualitative difference in the displacement of GAPM1a-depleted parasites could be observed by video microscopy (Supplementary Movies 2 and 3). However, the ability of GAPM1a-deficient parasites to invade host cells was not compromised (Fig. 3b). Although subtle defects may be missed by these assays[39], our observations suggest that GAPM1a is not essential for 2D motility or invasion.

We examined parasite egress by monitoring infiltration of the membrane-impermeable nucleic-acid stain DAPI into host cells over time, after zaprinast-stimulated egress[40]. GAPM1a-AID depletion did not affect egress kinetics, even after overnight treatments where all vacuoles showed profound morphological abnormalities (Fig. 3c). To confirm this unexpected result, we monitored zaprinast-induced egress by video microscopy of the parental strain and GAPM1a-AID untreated or treated with IAA for 18 h. Both the parental (Supplementary Movie 4) and untreated GAPM1a-AID (Supplementary Movie 5) strains egressed normally and invaded adjacent cells. After 18 h of IAA treatment, GAPM1a-AID vacuoles lysed the host cells and released the parasite masses from their bounding membranes (Supplementary Movie 6 and Fig. 3d). However, these abnormal parasites were unable to glide or invade host cells. This demonstrates that after 18 h of IAA treatment, GAPM1-AID parasites remain capable of secreting micronemes and lysing host cell membranes.

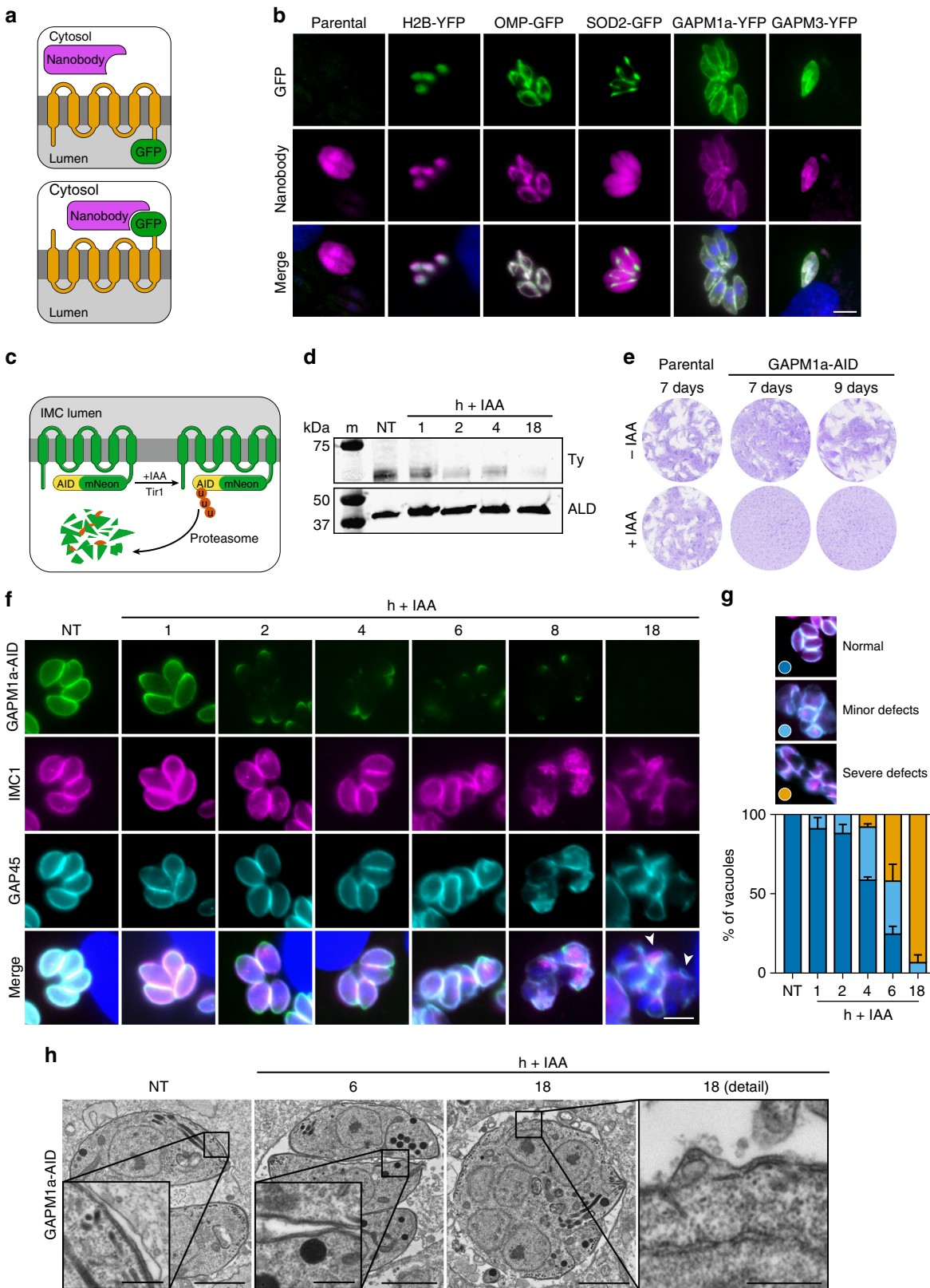

**Mature cortical microtubule arrangement depends on GAPM1a.** While analysing the localization of GAPM1a-AID by SR-SIM during its depletion, we observed that signal persisted along longitudinal paths reminiscent of cortical microtubules. To determine whether GAPM1a associates with the parasite cytos-keleton, we stably expressed mTagRFP_TgTubA1 in the

conditional mutant (GAPM1a-AID/mTagRFP-T_TgTubA1). We observed that GAPM1a-AID persisted along the path of cortical microtubules during depletion and was primarily retained at the apical end of these structures following IAA treatment (Fig. 4a). In rare cases, co-localisation of GAPM1a-AID with microtubules could be seen in untreated cells (Fig. 4a, first panel), although it

**Fig. 2** Conditional depletion of GAPM1a results in severe replication defects. **a** Strategy to experimentally determine protein topology. **b** Localisation of the mCherry-labelled GFP nanobody (GFPnano) in strains expressing different GFP/YFP-tagged proteins. The nanobody relocalises to sites of cytosolic GFP (H2B-GFP and OMP-GFP), but not to the luminal GFP of SOD2-GFP, as expected. Relocalisation of the nanobody in parasites expressing either GAPM1a-YFP or GAPM3-YFP indicates that the C-termini of these proteins are cytosolic. **c** Diagram of the auxin-induced degron (AID) system used to regulate GAPM1a expression. **d** Immunoblot showing rapid GAPM1a-AID depletion following treatment with IAA. GAPM1a-AID was visualised with an anti-Ty antibody, and anti-aldolase used as a loading control. Not treated (NT). **e** GAPM1a-AID depletion blocks plaque formation. Because of a reduction in plaque size for the untreated GAPM1a-AID strain, plaques were observed at 7- and 9-days post infection. Results are representative of three independent experiments. **f** Time course of morphological changes following the GAPM1a depletion after the addition of IAA. Parasites were stained with anti-GAP45 (cyan) and anti-IMC1 (magenta) antibodies and DNA visualised by Hoechst staining (blue). **g** Morphological changes were quantified in treated GAPM1A-AID parasites. After 4 h of IAA treatment, almost 50 % of vacuoles displayed morphological abnormalities, and by 18 h there were no normal vacuoles. Over 100 vacuoles were scored at each timepoint. Graph represents mean ± SD for $n = 2$ independent experiments. **h** TEM images of GAPM1a-AID parasites untreated or treated for 6 or 18 h with IAA. Boxes show detailed view of the IMC. Scale bars are 2 μm, and 500 nm within the detail. Unspecified scale bars are 5 μm. Source data are provided as a Source Data file

was usually evenly distributed throughout the IMC. To determine whether GAPM1a was sufficiently abundant to homogeneously cover the IMC membrane or more likely dispersed in a pattern below the limit of resolution, we counted the number of molecules using fluorescently labelled virus particles as a calibration standard (Supplementary Figure 3a–b)[41]. Calculating the two-dimensional distribution of mNeonGreen-tagged GAPM1a strains, we estimate $903 \pm 199.5$ (SD, $n = 305$) molecules μm⁻² for the direct fusion of GAPM1a and the fluorophore and $982 \pm 207$ (SD, $n = 278$) molecules μm⁻² for GAPM1a-AID (Supplementary Figure 3c). As expected, the abundance of GAPM1a-AID fell to $98 \pm 42$ (SD, $n = 132$) molecules μm⁻² when the parasites were treated with IAA for 4 h. The estimated abundance of GAPM1a in the IMC membrane is therefore significantly below the concentration for a crystalline array and consistent with a more disperse periodic arrangement below the limit of resolution.

We investigated whether loss of GAPM1a affected cortical microtubules. After 4 h of IAA treatment, parasites lacking GAPM1a display aberrant cortical microtubule arrays (Fig. 4b). Prolonged depletion of GAPM1a, caused a complete loss of normal cortical microtubules (open arrowhead, Fig. 4b), although relatively normal daughter microtubule scaffolds were occasionally observed. (asterisks, Fig. 4b). Loss of cortical microtubules and defects in their arrangement were also seen upon depletion of GAPM2a-AID (Supplementary Figure 2c). The prevalence of such defects increased over time. After 4 h of IAA treatment, approximately half of the parasites had abnormal cortical microtubules, and normal maternal arrays were absent after 18 h of treatment (Fig. 4c).

We confirmed that the impact of GAPM1a loss could also be observed in extracellular parasites. GAPM1a-AID parasites were treated extracellularly with IAA for 2 or 4 h. After 2 h of treatment, microtubules appeared largely normal; however, by 4 h of treatment, normal cortical microtubule arrays were rare (Fig. 4d). These results demonstrate that the stability of cortical microtubules depends on the presence of GAPM1a.

We characterized the changes in cortical microtubules following GAPM1a degradation, using live SR-SIM. Gaps in the cortical microtubule array could be observed as early as 2 h after the addition of IAA (arrows, Fig. 4e). To quantify this phenotype, we measured the distance between neighbouring microtubules. Since maternal microtubules are known to dissolve at the conclusion of endodyogeny, cells that contained daughter microtubule arrays were excluded. In the untreated control, the mean distance between microtubules was $362 \text{ nm} \pm 95 \text{ nm}$ (SD, Fig. 4f), comparable to the expected 322 nm spacing for the 22 cortical microtubules (based on a 2.26 μm cell diameter, measured from TEM sections of 20 untreated cells). After 2 h of GAPM1a depletion, the spacing of microtubules had significantly changed, with larger gaps appearing more frequently. Such gaps were more

evident after 4 h of IAA treatment, when ~40 % of cells lacked cortical microtubules (Fig. 4e, f).

Microtubules less than ~150 nm apart cannot be resolved by SR-SIM. We therefore analysed coronal TEM sections at the level of the nucleus of untreated or IAA-treated parasites, since GAPM1a depletion was first observed in this region of the cell. In untreated parasites, microtubules were observed close to the IMC in regularly spaced arrays (arrows, Fig. 4g). In contrast, after 6 h of IAA treatment, microtubules appeared clustered with large intervening spaces (Fig. 4g). We measured the spacing between clearly visible microtubules in these TEM sections. The microtubules from untreated samples were regularly arrayed at a distance of $298 \pm 60$ nm (SD). In contrast, the microtubules of GAPM1a-depleted parasites showed a broader range of distributions from 2–676 nm with a mean $256 \pm 121$ nm (SD), which differed significantly from the untreated distribution. The bundling observed in ~20% of treated parasites would not have been resolved by SR-SIM, but likely represents the zones of increased fluorescence (Fig. 4e). The precise positioning of cortical microtubules therefore depends on GAPM1a.

**Parasite morphology is altered by acute GAPM1a-AID depletion.** Chemical or genetic perturbation of parasite microtubules, like those observed upon GAPM1a depletion, can alter the size and shape of parasites[22,42]. During GAPM1a depletion, we observed changes in vacuole morphology, which we analysed by live microscopy. An hour after the addition of IAA, we observed that vacuoles lost their characteristic rosette arrangement, leading to an initial increase (60–100 min) and subsequent contraction (120–160 min) in the area occupied by each vacuole (Fig. 5a, Supplementary Movies 7 and 8). Following similar kinetics, the mean length of IAA-treated parasites decreased by ~15% compared to untreated controls and remained lower for the remainder of the experiment (Fig. 5b, Supplementary Movies 9 and 10). To determine whether extracellular depletion of GAPM1a similarly altered cell shape, we measured the length and circularity by automated analysis of parasites stained with an anti-SAG1 (Supplementary Figure 4a). After 2 h of IAA treatment there was a significant decrease in mean length ($p = 0.0007$, $n = 360$ untreated, 324 treated, Student's $t$ test) and an increase in circularity ($p < 0.0001$, $t$ test, $n = 404$ untreated, 224 treated, Supplementary Figure 4b). Such changes were more pronounced after 4 h of IAA treatment and mirrored the changes in length and circularity observed when GAPM1a was depleted for similar times before mechanically releasing parasites from host cells (Supplementary Figure 4a and b). These observations show that depletion of GAPM1a-AID leads to rapid changes in cell length and in the forces that maintain the shape of the vacuole, independently of parasite replication.

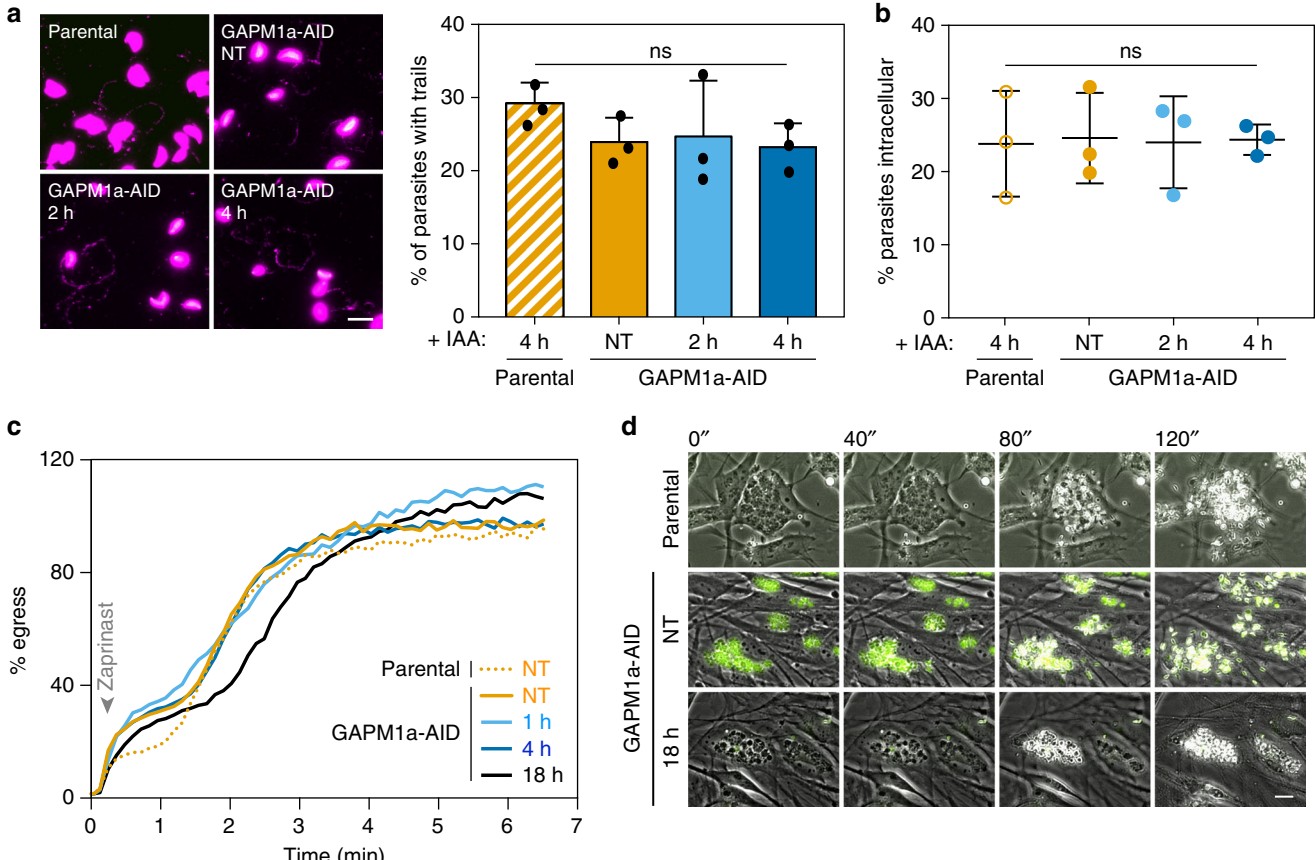

**Fig. 3** Acute depletion of GAPM1a does not inhibit 2D gliding, invasion, or egress. **a** 2D gliding motility assayed by trail deposition, visualised with anti-SAG1 using the parental line or GAPM1a-AID parasites after treatment with IAA for 2 or 4 h. Graph represents mean ± SD for $n = 3$ independent experiments; ns $p > 0.05$, Student's $t$ test. Scale bar is 10 μm. **b** Depletion of GAPM1a for up to 4 h had no effect on invasion. All IAA treatments were performed on intracellular parasites, prior to mechanical release and incubation with fresh monolayers for 1 h. The percentage of parasites that were intracellular was quantified. Each point represents the average invasion from one well. Graph represents mean ± SD for $n = 3$ independent experiments; ns $p > 0.05$, Student's $t$ test. **c** Zaprinast-induced egress measured as the percent of host cells permeabilized over time. The number of DAPI-positive host-cell nuclei was calculated by automated image analysis. Results are the mean of three wells and representative of two independent experiments. **d** Video microscopy of zaprinast-induced egress; representative frames are shown. Parasites depleted of GAPM1a-AID overnight lysed the host cells, despite remaining immobile and unable to reinvade for the course of the experiment. Scale bars are 10 μm. Source data are provided as a Source Data file

To compare the morphological effects of GAPM1a loss to other triggers of microtubule destabilization, we used the $\Delta TLAP2\Delta SPM1\Delta TLAP3$ parasite line (TKO) in which microtubules depolymerise after 4 h of incubation at 4 °C (Supplementary Figure 5)[13]. The length of TKO parasites was measured under conditions that maintain or depolymerise the cortical microtubule arrays. Untreated GAPM1a-AID parasites incubated extracellularly at 37 or 4 °C for 4 h did not change in length, although incubation with IAA for 4 h at 37 °C did result in shorter cells (Fig. 5c). The decrease in parasite length upon GAPM1a depletion was comparable to the effect of depolymerising microtubules by incubating the TKO strain at 4 °C (Fig. 5c).

The changes in parasite length led us to investigate whether other morphological parameters were altered by the loss of GAPM1a. We measured parasite volume using a Coulter counter. We saw a decrease in mean cell volume when GAPM1a-AID parasites were treated with IAA for 4 h intracellularly or when microtubules were depolymerised in the TKO line by extracellular incubation at 4 °C for 4 h (Fig. 5d). As a control, incubating the untreated GAPM1a-AID parasites at 37 or 4 °C for 4 h did not change their volume. Next, we determined parasite density using a suspended microchannel resonator (SMR) previously developed for yeast and mammalian cells[43,44], which allows the precise

quantification of buoyant mass. There was no significant change in buoyant mass after intracellular depletion of GAPM1a, while the TKO strain exhibited a small decrease in buoyant mass upon microtubule depolymerisation (Fig. 5e). Since cell density can be described by buoyant mass over volume[45], we infer that loss of microtubules through GAPM1a depletion or microtubule depolymerisation results in increased cell density. This shows that parasites become smaller, rounder, and denser without the support of the cortical microtubule cytoskeleton.

We examined the effect of GAPM1a depletion on 3D motility, since mutations that alter parasite shape have been previously linked to severe defects in 3D gliding through Matrigel, despite having little impact on 2D motility[39]. GAPM1a-AID parasites cultured in the presence or absence of IAA for 4 h were mechanically released and placed in a Matrigel chamber, where they were imaged by collecting z-stacks every second for a minute. Motility was tracked by the position of the Hoechst-stained nuclei of live parasites. Untreated GAPM1a-AID parasites moved in the corkscrew fashion previously observed for wild-type parasites; upon depletion of GAPM1a-AID, parasites were unable to glide through the matrix (Fig. 5f) and displayed a decreased displacement (Fig. 5g).

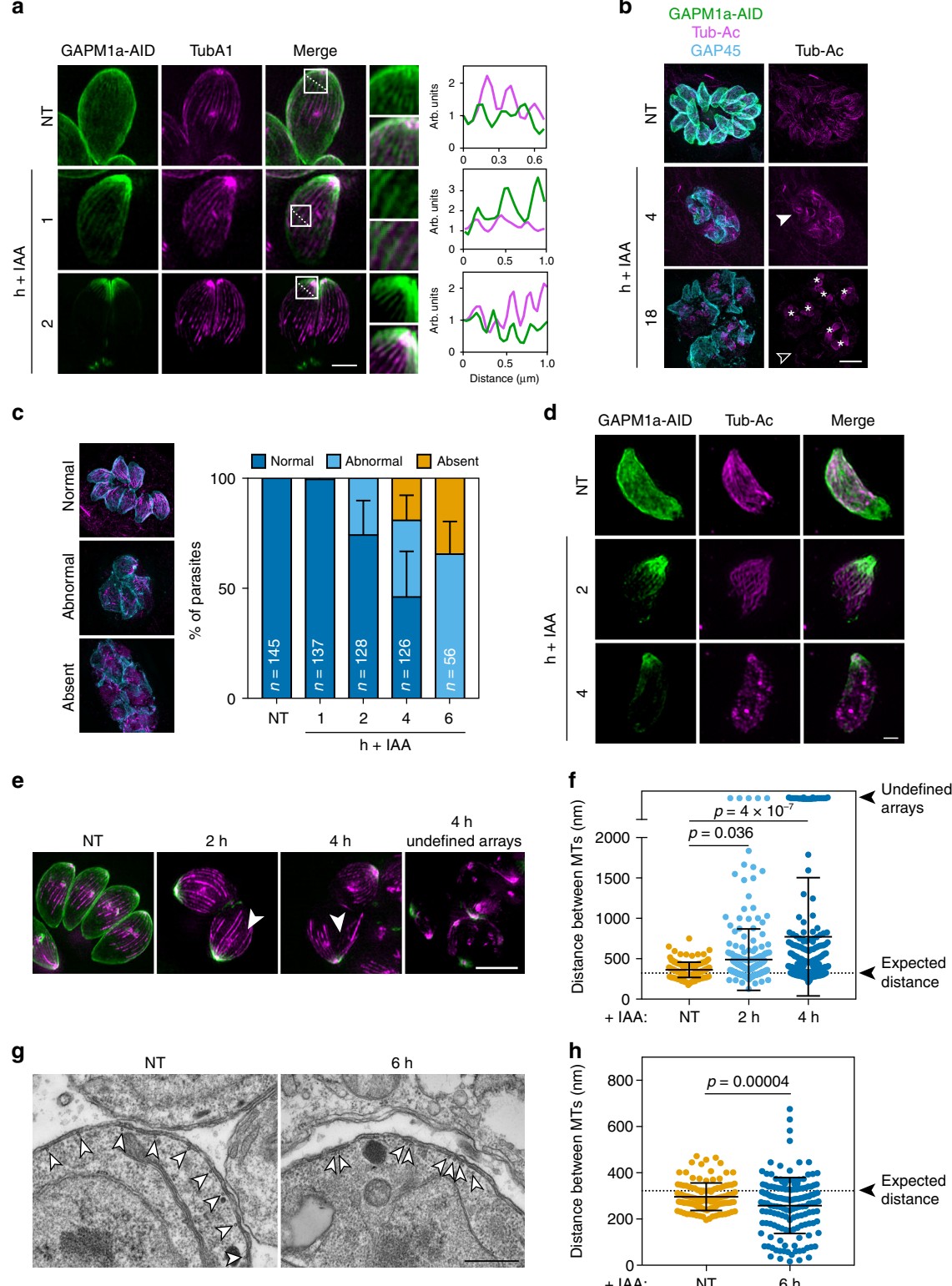

## Discussion

Apicomplexans have elaborate and unusually stable cortical microtubules that are unlike the highly dynamic cytoskeletal structures that typically support mammalian cell shape. It is believed that interactions between the alveoli, SPN, and cortical microtubules are required to establish and maintain the shape and rigidity of parasites. However, direct evidence for the function of these associations has been lacking. Here, we examine the

role of the GAPM protein family from which three members (GAPM1a, GAPM2a, and GAPM3) are highly expressed in *T. gondii* asexual stages, localise to the IMC, and are predicted to be essential. We demonstrate that the C terminus of GAPM1a is exposed to the cytosol, enabling rapid and conditional degradation of the protein using the AID system to analyse the effect of its depletion. We show that acute GAPM1a loss does not prevent 2D gliding motility, invasion, or induced egress; however, its

**Fig. 4** Depletion of GAPM1a results in cortical microtubule defects. **a** Colocalization of GAPM1a-AID (green) and cortical microtubules (mTagRFP-T_TubA1, magenta) by live SR-SIM following different treatments with IAA or vehicle (NT). Dotted lines indicate path of the profile plotted for the relative intensity of GAPM1a-AID (green) and mTagRFP_TubA1 (magenta). Scale bar is 5 μm. **b** GAPM1a-AID parasites treated for the indicated time with IAA were fixed and stained for acetylated tubulin (Tub-Ac, magenta) and GAP45 (cyan); GAPM1a-AID fluorescence is visible in green. Aberrant cortical microtubule arrangements (closed arrowhead) and depolymerisation (open arrowhead) are indicated. Daughter cell cortical arrays observed after 18 h of IAA treatment (asterisks) are highlighted. Scale bar is 5 μm. **c** Prevalence of cortical microtubule defects quantified over time from SR-SIM images. Graph represents mean ± SD for $n$ = number of parasites indicated on each bar. **d** Acetylated microtubules (magenta) in extracellular parasites treated with IAA for 2 or 4 h prior to fixing and staining. Images are representative of two independent experiments. **e** Live SR-SIM of non-treated (NT) and IAA-treated GAPM1a-AID/ mTagRFP-T_TubA1 parasites. Gaps in the microtubule array are indicated (arrows). Scale bar is 5 μm. **f** Distance between adjacent microtubules in **e**. Each dot represents the distance measured between adjacent microtubules from at least 50 cells in three independent experiments. Graph represents mean ± SD for $n$ = 137 (NT), 279 (2 h), 175 (4 h) measurements; $p$ values from two-sample Kolmogorov-Smirnov (KS) test. **g** Coronal TEM sections showing the arrangement of cortical microtubules (arrowheads) following treatment with vehicle or IAA for 6 h. Scale bar is 500 nm. **h** Distance between cortical microtubules observed in **g**. Each dot represents a single distance measured as above from two independent experiments. Graph represents the mean ± SD for $n$ = 124 (NT), 169 (6 h); $p$ value from two-sample KS test. Source data are provided as a Source Data file

absence during replication triggers severe morphological defects, similar to the loss of other IMC structural components. GAPM1a functions to maintain the shape of the parasite cell, and its depletion causes a rapid reduction in cell length and disrupts the arrangement of parasites within the vacuole. The morphological changes are related to the disorganisation and destabilisation of the cortical microtubules, which disrupt parasite 3D motility and precede defects in replication. These observations establish GAPM proteins as essential links between the cortical microtubules and the alveoli, necessary for the stability of apicomplexan cytoskeletons.

By endogenously tagging the five members of the GAPM family, we showed that all are expressed and localised to the IMC. However, GAPM1b and GAPM2b are barely detectable and appear to be dispensable in culture. Aside from their general distribution along the entire IMC, GAPMs were concentrated at unusual ring-like structures. Although we cannot rule these structures out as artefacts of fluorescent tagging, they resemble micropores previously visualized by electron microscopy[34]. Transiently observed dually tagged strains showed that pairs of GAPMs colocalized to these structures, although such strains could not be maintained in culture presumably due to the cumulative deleterious effects of tagging. GAPMs may nonetheless serve as markers for the future characterization of such putative micropores.

The orientation of all GAPMs appears to leave their C termini exposed to the cytosol. Using the high-affinity interaction between YFP and an anti-GFP nanobody[35] expressed in the parasite cytosol, we confirmed this predicted topology for both GAPM1a and GAPM3, which is further supported by the accessibility of the AID tags for degradation of GAPM1a and GAPM2a. This nanobody-based method of determining protein topology provided an unambiguous assay that can be easily used to probe the orientation of tagged protein termini for other parasite proteins by live-cell microscopy.

We used the AID system[30,38] to achieve rapid, conditional depletion of GAPM1a and GAPM2a. This allowed us to determine the impact of acute depletion of these proteins on cellular processes, which are otherwise obscured by the severe morphological defects caused by IMC perturbation during replication. Acute GAPM1a depletion did not grossly alter 2D gliding motility, invasion, or egress. Instead, loss of GAPM1a caused the rapid disordering and eventual depolymerisation of cortical microtubules. Depletion of GAPM1a in extracellular parasites caused similar cytoskeletal defects, further emphasising that these changes are independent of replication and are the direct effect of GAPM1a loss. Once formed, *T. gondii* cortical microtubules are extraordinarily stable and resist detergent extraction and cold

shock without depolymerisation[10,46]. Deletion of SPM1 increases sensitivity of cortical microtubules to detergents[12], and deletion of two additional MAPs ($\Delta TLAP2\Delta SPM1\Delta TLAP3$, TKO) renders them sensitive to cold-induced depolymerization[11]. None of the single MAP deletions performed to date dramatically affect cortical microtubules under normal growth conditions, although the TKO has a growth defect and occasionally results in the formation of aberrant parasites[11]. It is therefore surprising that depletion of single members of the GAPM family is sufficient to alter the cortical microtubule arrangement and stability. Based on biochemical evidence of their interaction[30], their enrichment at similar structures, and their lack of redundancy, we suggest that GAPM1a, GAPM2a and GAPM3 are all essential components of the same functional complex.

Since the first studies of apicomplexan ultrastructure, the ordered and tight association of cortical microtubules has been appreciated[5,47]. Here we show that this arrangement requires the GAPM proteins. Based on their localisation, topology, and persistence along the length of the microtubules during depletion, we propose that GAPM proteins act as a tether linking the cortical microtubules to the alveolar membrane. However, it is unlikely that GAPM proteins interact directly with tubulin, as cortical microtubules are heavily decorated with MAPs[10,11,48]. Freeze-fracture and cryo-electron tomography have identified inner membranous particles (IMPs) on the alveoli that follow the periodicity of the MAPs[8,10], as well as linkers that appear to bridge the space between the microtubules and the alveolar membrane[14]. Both IMPs and linkers extend along the entire length of the parasite, suggesting that the arrangement of the cortical microtubules may be established by proteins within the alveoli, and our work suggests GAPMs as likely components of these structures. Based on calibrated fluorescent microscopy, we calculate that GAPM1a is present at 903–982 copies per μm², remarkably similar to the distribution of IMPs within the IMC (~980 particles per μm² based on the 32 nm periodicity of the IMP lattice[10]). We posit that the interaction between the cortical microtubules and the alveoli provides essential stability to these cytoskeletal structures. However, the ability of cortical microtubules to withstand detergent extraction[10,11] demonstrates their intrinsic stability in the absence of cellular components or membranes. It is therefore likely that the function of GAPM1a is to prevent microtubule depolymerization by cellular factors that are absent from the extracted cytoskeletons, and that the specific endowment of cortical microtubules with such properties depends on their GAPM1a-mediated association with alveolar membranes. Physical interactions between microtubules and membranes have been observed in a wide range of organisms from trypanosomes[3,49] to plants[50–52]. In plants, transmembrane

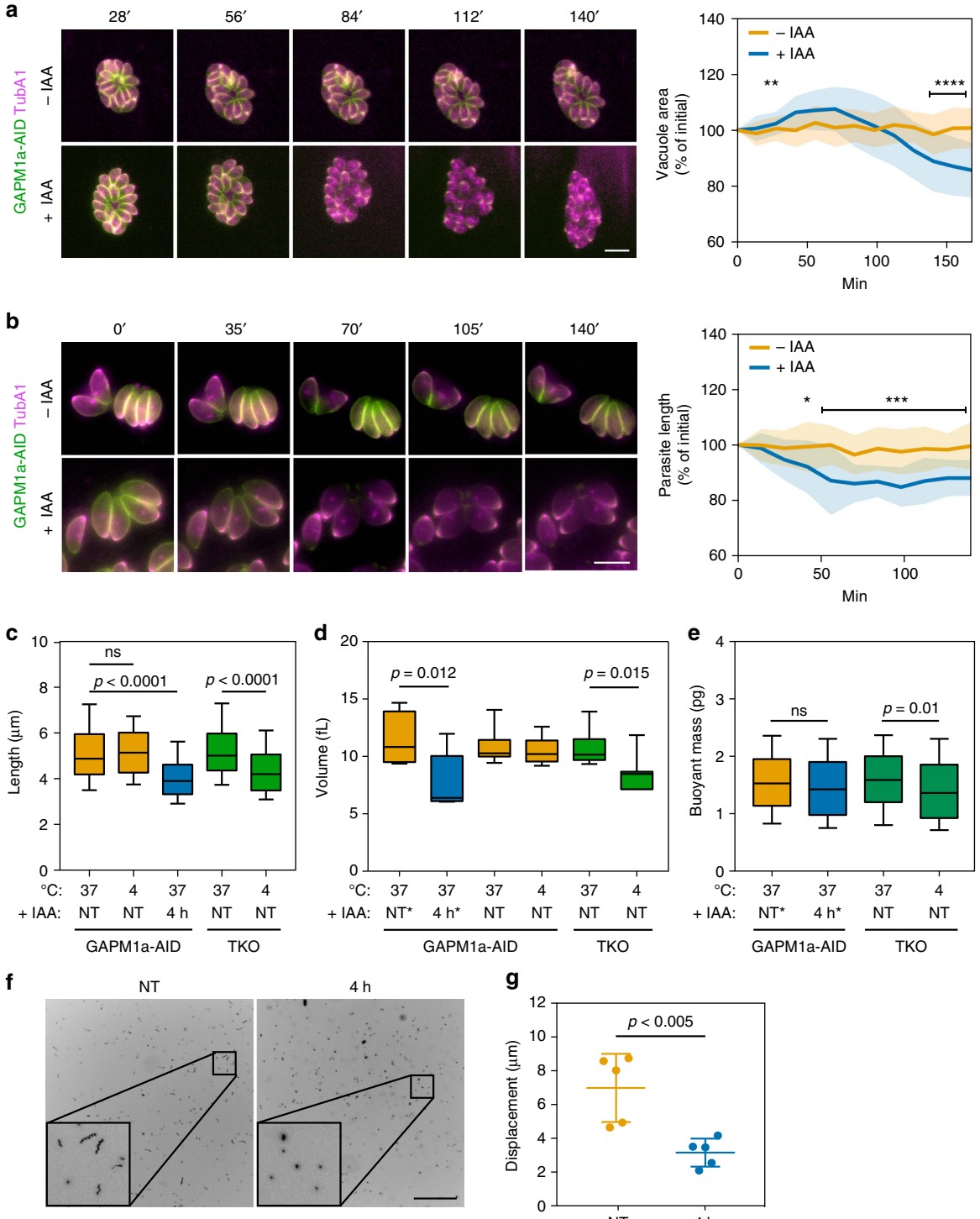

proteins are also suspected to tether and stabilize microtubules at the plasma membrane[50], suggesting convergent strategies for establishing cortical rigidity.

The defects in the cortical cytoskeleton caused by GAPM1a depletion triggered profound changes in cell shape and rigidity. Video microscopy revealed the disorganization of GAPM1a-deficient parasites within the vacuole caused by an apparent loss of their rigidity. Individual parasites became shorter and rounder upon GAPM1a depletion, and these morphological changes were mimicked by the depolymerisation of microtubules following cold shock of the TKO strain. Disrupting cortical microtubules by

either method also led to significant changes in cell volume while maintaining a constant buoyant mass, which indicates that the cortical cytoskeleton exerts outward pressure on the plasma membrane. Consistent with such a structural role, we found a correlation ($r^2 = 0.878$) between the number of microtubules and the approximate surface area of apicomplexan zoites (Fig. 6). This relationship suggests that the cortical microtubule cytoskeleton scales with the size and shape of cells in this phylum.

Recent studies have implicated the IMC protein PhIL1 in zoite shape[22,53,54]. In *Plasmodium* spp. it has been shown that PhIL1 interacts with GAPMs and is important for cell morphology and

**Fig. 5** Depletion of GAPM1a causes rapid changes in vacuolar organisation, cell morphology, and 3D motility. **a** Live video microscopy of intracellular parasites co-expressing GAPM1a-AID and mTagRFP-T_TubA1. Acute depletion of GAPM1a-AID led to the loss of the characteristic rosette organisation. Graph represents mean ± SD for $n = 16$ vacuoles per condition; **$p < 0.005$, ****$p < 0.00005$, Student's $t$ test with Holm–Sidak correction. **b** Length of individual, non-dividing parasites was measured every 14 min. Graph represents mean ± SD for $n = 21$ parasites per condition, *$p < 0.05$, ***$p < 0.0005$, Student's $t$ test with Holm–Sidak correction. Representative images shown for (**a**, **b**). Scale bars are 5 μm. **c** Comparison of parasite length following extracellular GAPM1a depletion or microtubule depolymerisation after incubating TKO parasites for 4 h at 4° C. Box plots for $n = 561$ (NT, 37 °C), 348 (NT, 4°C), 326 (4 h), 616 (TKO, 37 °C), 266 (TKO 4 °C) parasites per condition aggregated from three independent experiments. **d** Parasite volume estimated from over 8000 parasites per sample by Coulter counter. Asterisk indicates treatments performed before mechanical release of parasites from host cells. Box plots for $n = 6$ (37 °C, 4 h*), 7 (37 °C, NT*), 8 (37 °C, NT and 37 °C,TKO), 7 (4 °C, TKO) biological replicates; $p$ values from two-tailed Student's $t$ test. **e** Buoyant mass of parasites was quantified using SMR. Box plots for $n = 479$ (NT), 509 (4 h), 442 (TKO 37ºC), 455 (TKO 4ºC) aggregated from three independent experiments; $p$ values from two-tailed Student's $t$ test. **f** Maximum intensity projections of GAPM1a-AID parasite 3D motility either untreated or pre-treated for 4 h with IAA. Scale bar is 100 μm. **g** Mean displacement measured for at least 150 parasite per experiment. Graph shows mean ± SD for $n = 5$ biological replicates; $p$ values from two-tailed Student's $t$ test. All box plots represent median and 25th and 75th percentiles and whiskers are at 10th and 90th percentiles. Source data are provided as a Source Data file

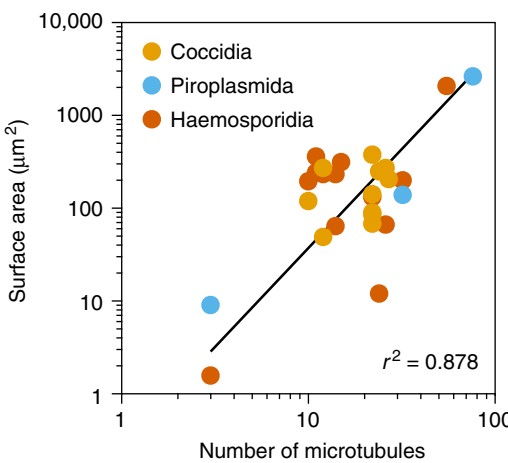

**Fig. 6** The number of cortical microtubules is correlated to parasite size. Analysis of the number of cortical microtubules and surface area of various apicomplexan species and life cycle stages from the published literature. Values and references shown in Supplementary Table 1

gametogenesis[53,54]. Downregulation of PhIL1 in *P. falciparum* blocks the extension of the IMC and its correct alignment with the extending cortical microtubules[53]. In developing gametocytes, PhIL1 localisation resembles that of GAPM1a at early stages of depletion[53]. In *T. gondii* tachyzoites, PhIL1 is not essential but deletion decreases fitness and results in shorter parasites[22]. Taken together, these data suggest that the GAPM proteins may interact with cortical microtubules partially through PhIL1 in both *T. gondii* zoites and during *P. falciparum* gametogenesis. Interestingly, disruption of PhIL1 caused subtle defects in 3D motility[39], which were not detected in 2D motility assays. GAPM1a depletion similarly results in a significant defect in 3D motility. We do not believe that this defect is due to a direct effect on the glideosome, as the parasites move largely normally in 2D. Rather, the alterations in cell shape caused by GAPM1a loss, and possibly the loss of cortical microtubules, result in impaired 3D motility, where the efficient coordination of forces along a longitudinal axis is likely more important.

The IMC provides an essential scaffold for daughter cell construction. We and others have shown that loss of structural IMC components results in severe morphological defects during parasite replication, underlining the importance of IMC composition during daughter cell formation[16,17,55]. The GAPMs, GAP40, and GAP50 are all seen at the developing IMC, suggesting that this core structural complex links the alveoli to the cortical microtubules at early stages of daughter cell assembly.

Since we observe daughter-cell cortical cytoskeletons in parasites depleted of GAPM1a, we suspect that microtubule polymerization and organisation, and the trafficking of alveolar membranes precede the stabilization of these structures by the GAPM proteins. Beyond its role in replication, the IMC also serves as a stable anchor for parasite motility. It is well appreciated that the glideosome is assembled in a two-step process. MyoA first binds its light chains (MLC1 and ELC1/2) and GAP45 forming a pre-complex in the cytosol[25,29] that is subsequently anchored to the IMC by GAP40 and GAP50 during daughter cell budding as the plasma membrane associates with the newly formed IMC[25,26]. It has been shown that disruption of MyoA, MLC1, or ELC have dramatic effects on parasite motility but do not affect parasite shape or replication[28,29,55]. In contrast, disruption of components of the structural complex (consisting of the cortical microtubules, GAPMs, GAP40, GAP50, and GAP45) results in dramatic defects in replication and misshapen parasites[16,28,56]. These results suggest the presence of functionally distinct portions of the glideosome—the motor portion involved in motility and the central structural core that serves to rivet cortical microtubules to the alveolar and plasma membranes.

Here we functionally characterise the roles of GAPM proteins in preserving the arrangement of microtubules and maintaining cell shape. We believe that GAPMs, along with GAP45, GAP40, and GAP50 define a core structural complex that links the plasmalemma and alveolar membranes directly to the cortical microtubules. These membrane interactions in turn impart stability to the cortical microtubules, whose arrangement maintains the cell shape of the zoite. Based on the conservation of the GAPM family, this critical role in cytoskeletal organization is likely conserved and critically important for the assembly and stability of similar structures throughout the apicomplexan phylum.

## Methods

**Parasites and host cells**. *T. gondii* tachyzoites from the strain RH and derived strains were maintained at 37 °C with 5% $CO_2$ growing in human foreskin fibroblasts (HFFs, ATCC SCRC-1041) cultured in Dulbecco's modified Eagle's medium (DMEM) supplemented with 10 or 3% heat-inactivated foetal bovine serum and 10 μg ml$^{-1}$ gentamicin. FlouroBrite DMEM (Thermo) with 10 % foetal bovine serum was used for live cell imaging.

**Plasmid generation**. For tagging the *GAPM* genes (with the exception of *GAPM2a*), the 3′ flank of the indicated gene, upstream of the stop codon, was amplified by PCR with primers as listed in Supplementary Table 2, inserted into pLIC-YFP-(3′-UTR$_{SAG1}$-pDHFR-HXGPRT-5′-UTR$_{DHFR}$)[57] by ligation-independent cloning[58], and linearized using a unique restriction site within the amplicon. The GFP-nanobody was amplified using primers P13 and P14 and inserted into the DD-myc-HXGPRT plasmid[36] using the restriction enzymes EcoRI/PstI. The fluorophore mCherry was amplified using P15 and P16 and inserted in frame between the DD-myc domain and GFP-nanobody using EcoRI.

Orientation of mCherry was checked by sequencing. The OMP-GFP line was created by amplifying OMP25 fused to eGFP[59] using primers P23 and P24 and cloning by Gibson assembly into *T. gondii* expression vector under the control of the ATP synthase β subunit promotor, amplified using P21 and P22. ISC3-3HA-CAT[33] (from Peter Bradley, UCLA) and mTagRFP-T_TubA1 (from Ke Hu, Indiana University)[60] were linearized using BglII prior to transfection.

**T. gondii strain generation**. Transfections of the parental Δ*KU80* line[58] were performed with a square-wave electroporator[61]. GAPM1a-YFP, GAPM1b-YFP, GAPM2b-YFP, and GAPM3-YFP strains were generated by transfecting 60 μg of each linearized tagging plasmid. The GAPM2a-mCherry line was created by transfection of 60 μg of plasmid containing Cas9 with the required sgRNA (sgRNA sequences are listed in Supplementary Table 2) along with 30 μg of purified PCR product using primers P17 and P18 into the Δ*KU80* line. The GAPM1a-AID and GAPM2a-AID lines were generated transfecting the sgRNA and Cas9 listed above into the Tir1 strain[38] along with the mNeonGreen-AID-Ty cassette amplified with homology arms using primers P9 and P10 (for *GAPM1a*) and P11 and P12 (for *GAPM2a*). Fluorescent parasites were isolated by FACS using an Aria II (BD Biosciences) at 48 h post-transfection and cloned by limiting dilution. The H2B-YFP[31] and SOD2-GFP[62] parasite lines have been previously described. Parasites transfected with the OMP-GFP plasmid were selected using 40 μM chloramphenicol.

**Invasion assay**. Mechanically released parasites were suspended in invasion media (DMEM supplemented with 1 % FBS, 20 mM HEPES, pH 7.4). $2 \times 10^5$ parasites per well were added to confluent HFF monolayers grown in 96-well plates and centrifuged at $290 \times g$ for 5 minutes. Invasion was allowed to proceed for 15 minutes at 37 °C, before the monolayers were fixed. Extracellular parasites were stained using mouse-anti-SAG1[31,63] (1:1000) conjugated to Alexa-Fluor-594 (Life Technologies), all parasites were stained with Alexa-Fluor-488-conjugated anti-SAG1 (1:100) after permeabilization with 0.25 % Triton X-100, and host cell nuclei were stained using Hoechst (Santa Cruz). Images were acquired using a Cytation3 imager (BioTek), and analysed using custom FIJI[64] macros to count the number of parasites and host-cell nuclei.

**Plaque formation**. Plaque assays were performed using the standard protocol[31]. Briefly, 500 parasites per well were inoculated into confluent HFF monolayers in 6-well plates, allowed to invade for an hour then treated with 500 μM IAA (used in all assays). At the indicated time, cells were fixed in ice-cold methanol for 10 minutes and stained with crystal violet (2 % crystal violet, 0.8 % ammonium oxalate, 20 % ethanol) for 10 minutes before washing with water, drying, and imaging.

**Quantitative egress**. Egress was quantified in a plate-based manner. Briefly, HFF monolayers in a clear bottomed 96-well plate infected with $5 \times 10^4$ parasites per well (MOI of 1) of parental or GAPM1a-AID for 24 h lines were treated with IAA as indicated. Before imaging, the media was exchanged for FlouroBrite DMEM supplemented with 10% IFS. Three images were taken before zaprinast (final concentration 500 μM) and DAPI (final concentration 5 ng mL$^{-1}$) were added, and imaging continued for 9 additional minutes before 1 % Triton X-100 was added to all wells to determine the total number of host cell nuclei. Results are the mean of three wells per condition and are representative of two independent experiments.

**Video microscopy of egress**. To capture egress, parental or GAPM1a-AID parasites were grown in HFFs in glass-bottom 35 mm dishes (MatTek Corp) at an approximate MOI of 1 with or without IAA for 18 h. Parasites were stimulated to egress with 500 μM zaprinast and recorded at 2–5 frames per second for 10 minutes using an Eclipse Ti microscope (Nikon) with an enclosure heated to 37 °C.

**2D gliding and cell shape**. Parasites were treated with IAA as indicated, mechanically released, resuspended in Ringers with 1 % IFS, and allowed to glide on poly-lysine coated coverslips for 15 minutes at 37 °C. Parasites were then fixed using 4 % formaldehyde (FA) and stained using mouse anti-SAG1 followed by goat anti-mouse-594. The number of parasites with trails was quantified using an Eclipse Ti microscope (Nikon)[65]. To determine cell shape, parasites were treated as above with IFS coated coverslips in place of poly-lysine. To examine extracellular length, parasites were mechanically released, filtered and incubated at 37 °C in intracellular buffer (142 mM KCl, 5 mM NaCl, 1 mM MgCl$_2$, 5.6 mM D-glucose, 2 mM EGTA, 25 mM HEPES, pH 7.4) for 4 h to prevent clumping. IAA was added at the indicated time before fixation and staining as above. Images were acquired as above, and cell length was calculated using custom FIJI macros to determine the maximum Ferret's diameter of each cell.

**Immunofluorescence microscopy**. Parasites were fixed with 4% FA for 10 minutes. Formaldehyde-fixed samples were permeabilised with 0.25% Triton X-100 in PBS and staining was performed with using anti-IMC1 (1:1000, Gary Ward, University of Vermont), anti-GAP45 (1:1000, Dominique Soldati-Favre, University of Geneva), anti-acetylated tubulin (1:1000, Sigma, T7451), and anti-HA (1:1000, Abcam, ab130275) primary antibodies and detected with Alexa-Fluor-labelled

secondary antibodies. Nuclei were stained with Hoechst (Santa Cruz) and coverslips were mounted in Prolong Diamond (Thermo Fisher). For live-cell microscopy, normal growth conditions were maintained throughout the experiment (37 °C, 5% CO$_2$), and images were taken at the indicated frequency. Super-resolution structured illumination microscopy (SR-SIM) was performed using a DeltaVision OMX microscope (GE Healthcare), heated to 37 °C for live cell imaging. FIJI and Imaris (Bitplane) software were used for image analysis and processing.

**Immunoblotting**. Prior to immunoblotting GAPM1a-AID parasites treated as indicated were suspended in lysis buffer (137 mM NaCl, 10 mM MgCl$_2$, 1 % Triton X-100, Halt protease inhibitors [Thermo Fisher], 20 mM HEPES pH 7.5). An equal volume of 2X Laemmli buffer (4 % SDS, 20 % glycerol, 5 % β-mercaptoethanol, 0.02% bromophenol blue, 120 mM Tris-HCl pH 6.8) was added, and the samples were heated to 65 °C for 10 minutes prior to separation of proteins by SDS-PAGE. After transferring separated proteins to nitrocellulose, membranes were blocked using 2 % milk in TBS-T (20 mM Tris-HCl, 138 mM NaCl, 0.1 % Tween-20, pH 7.5) before staining using mouse anti-Ty1[66] (1:2000) and rabbit anti-aldolase (1:2500, David Sibley, Washington University) with IRDye-680RD or IRDye-800RD (1:20,000, Li-Cor) secondary antibodies.

**Transmission Electron Microscopy**. For ultrastructural analyses, infected cells were fixed in a freshly prepared mixture of 1% glutaraldehyde (Polysciences Inc.) and 1 % osmium tetroxide (Polysciences Inc.) in 50 mM phosphate buffer at 4 °C for 45 min. The low osmolarity fixative was used to dilute soluble cytosolic proteins and enhance the visualization of cytoskeletal and conoid structure. Samples were then rinsed extensively in cold dH$_2$O prior to en bloc staining with 1 % aqueous uranyl acetate (Ted Pella Inc.) at 4 °C for 3 h. Following several rinses in dH$_2$O, samples were dehydrated in a graded series of ethanol and embedded in Eponate 12 resin (Ted Pella Inc.). Sections of 95 nm were cut with a Leica Ultracut UCT ultramicrotome (Leica Microsystems Inc., Bannockburn, IL), stained with uranyl acetate and lead citrate, and viewed on a JEOL 1200 EX transmission electron microscope (JEOL USA Inc.) equipped with an AMT 8 megapixel digital camera and AMT Image Capture Engine V602 software (Advanced Microscopy Techniques).

**3D motility assay**. GAPM1a-AID parasites treated intracellularly for 4 h with IAA or a vehicle control were mechanically released and resuspended to ~$1 \times 10^8$ parasite mL$^{-1}$ in prewarmed invasion media (DMEM supplemented with 1% FBS, 20 mM HEPES, pH 7.4) with 5 μg mL$^{-1}$ Hoechst 33342 (ThermoFischer Scientific). 50 μl of parasites were mixed with 40 μl of Matrigel (BD Biosciences). A concentration of 10 μl of this suspension was added to the Pitta chamber (constructed as described[39]) and polymerisation was allowed to proceed for 7 minutes at 27 °C. The slide was then moved to an Eclipse Ti microscope (Nikon) prewarmed to 34 °C and allowed to equilibrate. Using a triggered piezo stage, images were taken using a 10X objective with a 10 ms exposure. Z-stacks were obtained of 40 slices spaced 1 μm apart every second for 60 secs with $2 \times 2$ binning. Stacks were reconstructed and analysed with Imaris (Bitplane) image processing software using automated spot detection and tracking to export parameters of parasite motility.

**Cell mass, volume and density measurements**. To calculate the average density of a cell population, cell buoyant mass measured using a suspended microchannel resonator (SMR) was combined with cell volume measured by the commercial Coulter counter. SMR quantifies single-cell buoyant mass by a change in its resonant frequency as described in the literature[43]. The SMR devices were fabricated as using standard methods[43,44] by CEA-LETI, Grenoble, France based on a design that was described in a previous study[67]. A piezo-ceramic was placed under the device to resonate the SMR in its second flexural bending mode, and SMR resonant frequency was monitored using piezo-resistors embedded at the base of the cantilever[68].

Parasites treated as indicated were isolated from host cells and resuspended in DMEM supplemented with 10 mM HEPES buffer (pH 7.4) at ~$1 \times 10^7$ parasites ml$^{-1}$. 100 μl of this suspension was used to quantify volume by the Coulter counter. The remaining cells were loaded into the SMR from the inlet with a constant pressure controlled by the pressure regulators, achieving a typical flow rate of 2 nL s$^{-1}$. 300–500 cells were measured within 30 minutes at room temperature for each sample. The average density ($\rho$) of a cell population was calculated using the following equation:[45]

$$\rho = \rho_f + m_B/V, \qquad (1)$$

where $m_B$ is the median buoyant mass obtained from fitting a log-normal function to the buoyant mass distribution, and $V$ is the median cell volume from Coulter Counter.

**Surface area calculations**. Parasite surface area was estimated by calculating the surface area of an ellipsoid using the following equation:

$$S \approx 4\pi \left( \frac{(ab)^{1.6} + (ac)^{1.6} + (bc)^{1.6}}{3} \right)^{\frac{1}{1.6}} \tag{2}$$

where a, b and c represent the three axis of the ellipsoid.

**Protein number quantification**. Sindbis virions with TE12 tagged with mNeon[41] (provided by John Murray from Indiana University) were adsorbed onto at 35 mm glass-bottomed dish for 3–5 minutes at room temperature then immediately imaged using a 40X NA 1.4 lens and acquired using the Hamamatsu ORCA-ERA camera. Images were then flat-fielded and the background subtracted before automatic particle detection was used to measure the total intensity of viral particles. Over 700 virions were analysed and the photons ($p$) per virion per second calculated from the intensity using the following equation:[69]

$$p = \left[ \left( \frac{f}{(i_{max} - o)} \right) \times (i - o) \right], \tag{3}$$

where $f$ is the full well capacity of the detector, $imax$ is the maximum intensity value the detector can produce, $i$ is the intensity value being converted to photons, and $o$ is the detector offset. GAPM1a-mNeon parasites were allowed to infect an HFF monolayer for 2–4 h before extracellular parasites were washed away and the media exchanged for FlouroBrite DMEM containing 10% IFS. Images of live parasites were taken under the same exposure conditions, flat-fielded and background subtracted and regions of interest of between 1–2 μm$^2$ were manually selected on flat sections from fluorescent parasites and the intensity converted to photons per second per μm$^2$. The mean value of a fitted Gaussian curve was used to determine the photons per second from a single virion and used to estimate the number of labelled GAPM1a-mNeon proteins μm$^{-2}$.

**Reporting Summary**. Further information on experimental design is available in the Nature Research Reporting Summary linked to this article.

## Data availability:
The source data underlying Figs. 2g, 3a–c, 4c, 4f, 4h, 5a–e, and 5g, and Supplementary Figures 1d, 3a–c, and 4a–b is provided as a **Source Data** file. All other data is available from the corresponding authors by request.

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

## Acknowledgements

We would like to thank Peter Bradley, Gary Ward, Dominique Soldati-Favre, Ke Hu, and John Murray for kindly contributing reagents that made this study possible. Wendy Salmon assisted in quantifying protein numbers. Diego Huet generated the OMP-GFP parasite line, and Wandy Beatty (Washington University Molecular Microbiology Imaging Facility) performed the electron microscopy. This study was supported by a Sir Henry Wellcome fellowship to CRH (103972/Z/14/Z), an ERC research grant (ERC-2012-StG 309255-EndoTox) and Wellcome Senior fellowship (087582/Z/08/Z) to MM, an NIH Director's Early Independence Award (1DP5OD017892) and an NIH Exploratory R21 grant (1R21AI123746) to SL, and by the Institute for Collaborative Biotechnologies through a grant (W911NF-09-0001) from the US Army Research Office to SRM.

## Author contributions

CRH designed and performed experiments, analysed data, and wrote the paper; ES and MG designed reagents and methods used in this paper. JHK performed and analysed the mass and density measurements. SRM and MM supervised experimental design. SL designed experiments and wrote the paper.

## Additional information

**Competing interests:** The authors declare no competing interests.

