## [Peer Review File · Nature Communications]

Reviewers' Comments:

Reviewer #1:

Remarks to the Author:

The manuscript examines the localization, topological arrangement, and functional properties of a set of integral membrane proteins ("GAPM" proteins) of *Toxoplasma gondii*. The authors use structured illumination microscopy to show that the proteins are distributed over the entire alveolar membrane of the parasite with some concentration in small rings at the sutures joining adjacent alveolar sacs. A novel method employing intracellular expression of a nanobody, likely to be a broadly useful technique, was used to show that the extra-membranous portions of the GAPMs project into the cytosol. They go on to show that this topology renders the GAPMs vulnerable to targeted destruction using the auxin-induced degron (AID) approach. Important advantages of this approach are that significant destruction happens rapidly compared to the cell-cycle time, greatly reducing the problems of interpretation that result from intervening parasite replication, and the degradation can be induced even in parasites outside the host cell. The microtubule cytoskeleton of transgenic parasite lines expressing AID-tagged GAPMs, treated with indole acetic acid, becomes disordered within a few hours, and disappears entirely with prolonged treatment. Interestingly, the microtubules of developing daughter parasites are more resistant than those of the adult. This differential resistance to destabilizing factors is consistent with behavior observed when the microtubules were destabilized by knocking out three MT binding proteins (Liu et al. Mol Biol Cell 2016; citation #11 in the manuscript).

The disorganization of the microtubule cytoskeleton seems to have rather little effect on the parasites, but its eventual disappearance is accompanied by gross morphological abnormalities in the alveolar membranes and overall parasite shape. As observed with other conditions that disrupt the alveolar system, parasite replication is severely hindered by chronic knock-down of the three GAPMs previously shown to be "essential" by a CRISPR-based genome-wide screen (Sidik et al Cell 2016; citation #32 in the manuscript).

The work reported here is technologically excellent, entirely convincing, reasonably interpreted, and will be of immediate interest to the large number of scientists working with this phylum of notorious parasites. There is much to ponder here also for the wider community of cell biologists concerned with the organization of the cytoskeleton and internal membrane systems of all cells. The work should be published in Nature Communications. There are a few changes that might improve the manuscript, and a couple of questions that, if answerable, would aid interpretation..

For the wider audience of Nature Communications, this work contains information relevant to two broad themes that could be identified more clearly by these authors in their Discussion:

- (1) How do cells manage to endow different regions and pieces of their cytoskeleton with radically different properties, even though the different pieces are polymeric forms of nearly identical monomers and the regions are separated by diffusion times of the order of a few milliseconds? The various tubulin-based structures of *Toxoplasma gondii* present this conundrum in a particularly stark manner.

(2) What are the shared mechanisms, if any, used by cells to fashion and hold membranes into astonishingly diverse but reproducible morphologies, de novo with each cell division? Again, the various internal membrane systems of *Toxoplasma gondii*, wildly different from one another but precisely reproduced in every cell, surely provide a rich vein of answers, involving the GAPMs and other proteins.

Some more specific issues that merit the attention of the authors:

Discussion, page 9, lines 360-362;

“...however, none of the MAP deletions performed to date dramatically affect cortical microtubules under normal growth conditions.”

This overstates the case a bit. In the paper cited as reference 11, examples are shown of parasites grown normally at 37C that apparently completely lack cortical microtubules. This is an important detail, because those parasites maintain essentially normal morphology, suggesting that the link between cortical microtubules and alveolar morphology is more indirect than is suggested by the dramatic morphological changes accompanying microtubule disappearance observed after 18 hours of IAA treatment in the present work.

Results, subheading between lines 223 and 224

“GAPM1a is necessary for the positioning and stability of cortical microtubules”

Again referring to the paper cited as reference 11, the observation of parasites without microtubules shows that, while GAPM1a may be necessary for the stability of adult (but possibly not daughter?) cortical microtubules within parasites, it is not sufficient.

Discussion, page 10, lines 374-375;

“...the maintenance of stable microtubules in these parasites requires tight attachment to the alveolar membrane.”

The sentence is correct, but potentially misleading. Most readers will not be aware that the qualifier “in these parasites” is critical, and is to be interpreted strictly as “inside these parasites” . The cortical microtubules of *T. gondii* are easily extracted intact from lysed fragmented parasites, and are stable indefinitely, warm or cold, over a wide range of ionic conditions, with no attachment to membranes of any sort (Morrissette et al J Cell Sci 1997; cited as reference #10; Hu et al J Cell Biol 2002; cited as reference #47; Hu et al PLoS Path 2006). It seems that the cortical microtubules become unstable when detached from the alveolar membrane only if they are inside intact parasites. Could it be that detachment somehow activates or enables the intracellular (enzymatic?) mechanism responsible for the dissolution of maternal cortical microtubules during parasite replication?

Questions that will occur to many readers:

(1) I was surprised to find no mention of observations following IAA washout. How reversible are the changes caused by auxin induced degradation of the GAPMs? If IAA is removed, does the cortical microtubule array reappear? Do the cells then revert to normal morphology and growth? These observations might yield important insight into the requirements for establishing an ordered cortical MT array, and should be mentioned if available.

2) In untreated cells, the GAPMs seem to be uniformly distributed over the entire alveolar membrane surface when detected by light microscopy. One wonders if this is actually the case, or merely a low-resolution view of a more structured arrangement (e.g., associating with filaments of the SPN). It would be interesting to estimate the number of molecules of GAPM per untreated cell (e.g., from the Western blots), and from that compute the average surface density of GAPM molecules. Are there enough GAPM molecules to account for the known array of intra-membrane particles? Enough to decorate all filaments of the SPN? As a rough reality check on the calculation, one might take a figure of ~20,000 molecules per square micron as a feasible upper limit, beyond which membrane proteins may form continuous 2-D crystals.

3) The authors measure the length, volume, and circularity of treated and untreated parasites with impressive precision. In the acute phase, before gross changes in parasite shape become evident, they find small, though statistically significant changes in those parameters. Given that the range of values in untreated parasites is comparable to the difference between treated and untreated, are these small changes biologically significant? If the authors have reason to believe that they are, those reason should be given.

Text inaccuracies:

Introduction, page 2, lines 25-27

“In addition to cortical microtubules (also known as subpellicular microtubules), the asexual stages of *T. gondii* harbour four other distinct microtubule structures: a spindle in replicating parasites, centrioles, a conoid, and intraconoid microtubules”.

The conoid of *T. gondii* contains tubulin, but not microtubules.

Results, page 6, lines 226-227; page 7, line 248; Materials and Methods, page 11, line 446; Legends for Figs 4 and 5

The construct denoted “TubA1-RFP” is almost certainly incorrectly named. It is most likely mTagRFP-T_TgTUBA1. The distinction is important, since this construct was used for SIM. “RFP” with no modifiers would normally indicate the mRFP originally introduced by the Tsien lab, with which SIM imaging of cortical microtubules as shown here would probably be impossible.

Reviewer #2:

Remarks to the Author:

The paper entitled ‘Alveolar proteins stabilize cortical microtubules in *Toxoplasma gondii*’ by Harding et al describes the function of a family of multipass transmembrane proteins termed ‘GAPM’ play in the *Toxoplasma gondii*. The authors follow on from previous work demonstrating that these proteins are

integral membrane proteins of the IMC and likely hold a particular orientation and most likely function in co-ordinating the stability and position of microtubules (MTs). To prove this the authors use elegant high resolution imaging and molecular genetic techniques and develop a novel method to probe membrane topology.

The work is of very high quality and well interpreted and furthermore, does an good job of dissecting out the function of the GAPM's away from what could be pleiotropic effects. Further, it applies a new technique to probe the membrane topology of proteins using a chromobody system which will, no-doubt, be used in the field (is this the first time such a technique has been used? If so, important to highlight this).

The only major criticism that I have is that all conclusions on localisation have been drawn using genetically-tagged proteins (ie mCherry or GFP). This is only a real concern as it is clear that the parasites do not like having the C-terminus of GAPM tampered with, as it confers a fitness cost, suggesting the protein function is somehow compromised. For example, genetic tagging could interfere with correct protein trafficking, thus any localisation studies should be interpreted with caution. For example, Figure 1D and E look at substructures of GAPM within parasites, which could quite possibly be artefacts due to protein tagging. I suggest that reporting this level of detail does not provide much benefit due to this possible caveat.

MINOR POINTS

Figure 1: What basis is the membrane topology predicted? It is stated that this is a prediction in the figure legend but is not discussed at all. This is important to discuss the rationale for this if the topology is predicted to be different between clades.

Figure 2: Presents functional data on GAPM1a and TEM images in panel 'H' are used to demonstrate defects in membrane morphology upon loss of this gene. I'm unsure of the usefulness of these images as by 18hours the parasites look really unwell and thus effects could be just pleiotropic. The zoom-in box only highlights one area of many that I can see where there is problems with membrane integrity.

Figure 4A: What is the utility of looking at localisation of GAPM in relation to MT upon depletion rather than just showing colocalization at steady state?

Figure 6: There is not enough data to support this model of GAPM. There is no functional data to suggest that GAPMs play a role in anchoring the glideosome (as explored here in Figure 3) and there is no information has to how GAPM's interact with the MT. I would remove this.

This work will be an excellent addition to our understanding of microtubule biology in this early branching eukaryote and thus will be of interest to biologists working on fundamental aspects of apicomplexan biology as well as investigators studying the cytoskeletal dynamics and function.

REVIEWER COMMENTS

Reviewer #1

The manuscript examines the localization, topological arrangement, and functional properties of a set of integral membrane proteins (“GAPM” proteins) of *Toxoplasma gondii*. The authors use structured illumination microscopy to show that the proteins are distributed over the entire alveolar membrane of the parasite with some concentration in small rings at the sutures joining adjacent alveolar sacs. A novel method employing intracellular expression of a nanobody, likely to be a broadly useful technique, was used to show that the extra-membranous portions of the GAPMs project into the cytosol. They go on to show that this topology renders the GAPMs vulnerable to targeted destruction using the auxin-induced degron (AID) approach. Important advantages of this approach are that significant destruction happens rapidly compared to the cell-cycle time, greatly reducing the problems of interpretation that result from intervening parasite replication, and the degradation can be induced even in parasites outside the host cell. The microtubule cytoskeleton of transgenic parasite lines expressing AID-tagged GAPMs, treated with indole acetic acid, becomes disordered within a few hours, and disappears entirely with prolonged treatment. Interestingly, the microtubules of developing daughter parasites are more resistant than those of the adult. This differential resistance to destabilizing factors is consistent with behavior observed when the microtubules were destabilized by knocking out three MT binding proteins (Liu et al. Mol Biol Cell 2016; citation #11 in the manuscript).

The disorganization of the microtubule cytoskeleton seems to have rather little effect on the parasites, but its eventual disappearance is accompanied by gross morphological abnormalities in the alveolar membranes and overall parasite shape. As observed with other conditions that disrupt the alveolar system, parasite replication is severely hindered by chronic knock-down of the three GAPMs previously shown to be “essential” by a CRISPR-based genome-wide screen (Sidik et al Cell 2016; citation #32 in the manuscript).

The work reported here is technologically excellent, entirely convincing, reasonably interpreted, and will be of immediate interest to the large number of scientists working with this phylum of notorious parasites. There is much to ponder here also for the wider community of cell biologists concerned with the organization of the cytoskeleton and internal membrane systems of all cells. The work should be published in Nature Communications. There are a few changes that might improve the manuscript, and a couple of questions that, if answerable, would aid interpretation.

We appreciate the reviewer’s kind observations and comments, which prompted us to include additional data that now addresses some of the remaining open questions.

For the wider audience of Nature Communications, this work contains information relevant to two broad themes that could be identified more clearly by these authors in their Discussion:

(1) How do cells manage to endow different regions and pieces of their cytoskeleton with radically different properties, even though the different pieces are polymeric forms of nearly identical monomers and the regions are separated by diffusion times of the order of a few milliseconds? The various tubulin-based structures of *Toxoplasma gondii* present this conundrum in a particularly stark manner.

The reviewer keenly identifies one of most challenging questions in the field, and likely the consequence of interactions between the site of microtubule polymerization and the proteins

associated with them. The GAPMs provide some insight into this question because their integral localization to the alveolar membranes restricts their function to those particular sites. We have included this possibility in the discussion stating, *“It is therefore likely that the function of GAPM1a is to prevent microtubule depolymerization by cellular factors that are absent from the extracted cytoskeletons, and that the specific endowment of cortical microtubules with such properties depends on their GAPM1a-mediated association with alveolar membranes.”* Further studies will be needed to determine the mechanisms of stabilization, patterning, and length determination for cortical microtubules.

(2) What are the shared mechanisms, if any, used by cells to fashion and hold membranes into astonishingly diverse but reproducible morphologies, *de novo* with each cell division? Again, the various internal membrane systems of *Toxoplasma gondii*, wildly different from one another but precisely reproduced in every cell, surely provide a rich vein of answers, involving the GAPMs and other proteins.

This is a very interesting question indeed. Because of the innovation of using rapid protein degradation to study GAPM proteins, the present study has primarily focused on the acute changes to the parasite cytoskeleton. *The de novo* formation of the complex apicomplexan cytoskeleton is likely a very dynamic process, which will require further work to be properly resolved. As noted above, we believe that having a membrane-associated stabilizer may help stabilize structures that require temporally restricted patterning. We therefore note in the discussion that, *“Since we observe daughter-cell cortical cytoskeletons in parasites depleted of GAPM1a, we suspect that microtubule polymerization and organisation, and the trafficking of alveolar membranes precede the stabilization of these structures by the GAPM proteins.”*

Some more specific issues that merit the attention of the authors:

Discussion, page 9, lines 360-362;

“...however, none of the MAP deletions performed to date dramatically affect cortical microtubules under normal growth conditions.”

This overstates the case a bit. In the paper cited as reference 11, examples are shown of parasites grown normally at 37C that apparently completely lack cortical microtubules. This is an important detail, because those parasites maintain essentially normal morphology, suggesting that the link between cortical microtubules and alveolar morphology is more indirect than is suggested by the dramatic morphological changes accompanying microtubule disappearance observed after 18 hours of IAA treatment in the present work.

We agree with the reviewer’s comments and have toned down the statement accordingly, which now reads, *“None of the single MAP deletions performed to date dramatically affect cortical microtubules under normal growth conditions, although the triple knockout has a growth defect and occasionally results in the formation of aberrant parasites.”* As indicated above, it is likely that the IMC is stabilized by different processes during and after its formation. It is therefore possible that some MAPs act later than the GAPMs, thereby differentially impacting endodyogeny and parasite morphology. In addition, the reference cited [11] did not report whether the cells lacking cortical microtubules eventually die or accrue gross morphological changes.

Results, subheading between lines 223 and 224

“GAPM1a is necessary for the positioning and stability of cortical microtubules”

Again, referring to the paper cited as reference 11, the observation of parasites without microtubules shows that, while GAPM1a may be necessary for the stability of adult (but possibly not daughter?) cortical microtubules within parasites, it is not sufficient.

We agree that depletion of GAPM1a does not appear to significantly affect the stability of daughter microtubules, although we cannot speak with much certainty on their precise positioning as they are below the detection limit for SIM. We have changed the subheading to more accurately reflect this limitation, it now reads, *“Positioning and stability of mature cortical microtubules depend on GAPM1a.”*

Discussion, page 10, lines 374-375;

“...the maintenance of stable microtubules in these parasites requires tight attachment to the alveolar membrane.”

The sentence is correct, but potentially misleading. Most readers will not be aware that the qualifier “in these parasites” is critical, and is to be interpreted strictly as “inside these parasites”. The cortical microtubules of *T. gondii* are easily extracted intact from lysed fragmented parasites, and are stable indefinitely, warm or cold, over a wide range of ionic conditions, with no attachment to membranes of any sort (Morrisette et al J Cell Sci 1997; cited as reference #10; Hu et al J Cell Biol 2002; cited as reference #47; Hu et al PLoS Path 2006). It seems that the cortical microtubules become unstable when detached from the alveolar membrane only if they are inside intact parasites. Could it be that detachment somehow activates or enables the intracellular (enzymatic?) mechanism responsible for the dissolution of maternal cortical microtubules during parasite replication?

The reviewer raises a very good point. We have modified the discussion to reference to the intrinsic stability of extracted microtubules and the possibility of their destabilization as an active process in live cells. Lines 1299-1304 now read, *“The ability of cortical microtubules to withstand detergent extraction^{10,11} demonstrates their intrinsic stability in the absence of cellular components or membranes. It is therefore likely that the function of GAPM1a is to prevent microtubule depolymerization by cellular factors that are absent from the extracted cytoskeletons, and that the specific endowment of cortical microtubules with such properties depends on their GAPM1a-mediated association with alveolar membranes.”*

Questions that will occur to many readers:

(1) I was surprised to find no mention of observations following IAA washout. How reversible are the changes caused by auxin induced degradation of the GAPMs? If IAA is removed, does the cortical microtubule array reappear? Do the cells then revert to normal morphology and growth? These observations might yield important insight into the requirements for establishing an ordered cortical MT array, and should be mentioned if available.

We did attempt washout experiments. However, there is no recovery of GAPM1a to the mature alveoli, and new protein is only deposited onto daughter cells. A similar situation has previously been documented for GAP40 using a photoactivatable fluorophore [Ouologuem and Roos 2014]. It therefore appears that there is little or no addition of newly synthesized integral membrane proteins to the mature alveoli, so we cannot readily examine the ability of newly synthesized

GAPM1a to rescue the microtubule defects we observe. An early IAA washout enables normal daughter cells to emerge; however, after 18 h of IAA treatment GAPM1a-positive daughter cells were never seen to emerge from the large parasite masses formed. This shows that there is a window of reversibility, however accumulation of morphological defects will eventually prevent the emergence of healthy cells.

2) In untreated cells, the GAPMs seem to be uniformly distributed over the entire alveolar membrane surface when detected by light microscopy. One wonders if this is actually the case, or merely a low-resolution view of a more structured arrangement (e.g., associating with filaments of the SPN). It would be interesting to estimate the number of molecules of GAPM per untreated cell (e.g., from the Western blots), and from that compute the average surface density of GAPM molecules. Are there enough GAPM molecules to account for the known array of intra-membrane particles? Enough to decorate all filaments of the SPN? As a rough reality check on the calculation, one might take a figure of ~20,000 molecules per square micron as a feasible upper limit, beyond which membrane proteins may form continuous 2-D crystals.

We agreed with the reviewer that estimating the density of GAPM1a molecules could be illuminating so we implemented calibrated fluorescence microscopy to determine GAPM1a-mNeonGreen molecules/ μm^2 as previously described [Murray 2017]. As we state in the discussion, “*we calculate that GAPM1a is present at 903–982 copies per μm^2 , remarkably similar to the distribution of IMPs within the IMC (~980 particles per μm^2 based on the 32 nm periodicity of the IMP lattice.*” While it is unlikely that single GAPM1a molecules constitute each IMP, we speculate that they contribute to these structures as heterotrimers with the other GAPMs, and possibly other IMC proteins.

3) The authors measure the length, volume, and circularity of treated and untreated parasites with impressive precision. In the acute phase, before gross changes in parasite shape become evident, they find small, though statistically significant changes in those parameters. Given that the range of values in untreated parasites is comparable to the difference between treated and untreated, are these small changes biologically significant? If the authors have reason to believe that they are, those reason should be given.

Indeed, we observe only relatively small changes in parasites morphology during extracellular or intracellular depletion of GAPM1a. Since similar changes have been shown to affect 3D motility [Leung 2014], we now include data investigating whether GAPM1a depletion affects this process (**Fig. 5f–g**). We found that depletion of GAPM1a-AID led to a significant block in 3D motility, although these parasites were able to move in 2D (**Fig. 2a**). We cannot formally distinguish whether the defect in 3D motility is exclusively caused by the changes in shape, so the discussion states, “*GAPM1a depletion similarly results in a significant defect in 3D motility. We do not believe that this defect is due to a direct effect on the glideosome, as the parasites move, largely normally, in 2D. Rather, the alterations in cell shape caused by GAPM1a loss—like those reported for PhIL1—and possibly the loss of cortical microtubules result in impaired 3D motility, where the efficient coordination of forces along a longitudinal axis is likely more important.*”

Text inaccuracies:

Introduction, page 2, lines 25-27

“In addition to cortical microtubules (also known as subpellicular microtubules), the asexual stages of T. gondii harbour four other distinct microtubule structures: a spindle in replicating parasites, centrioles, a conoid, and intraconoid microtubules”.

The conoid of *T. gondii* contains tubulin, but not microtubules.

Thank you, this oversight has been corrected in the text. The sentence now reads: *“In addition to cortical microtubules (also known as subpellicular microtubules), the asexual stages of T. gondii harbour four other distinct tubulin-containing structures: centrioles, a conoid, intraconoid microtubules, and a spindle in replicating parasites”*

Results, page 6, lines 226-227; page 7, line 248; Materials and Methods, page 11, line 446; Legends for Figs 4 and 5

The construct denoted “TubA1-RFP” is almost certainly incorrectly named. It is most likely mTagRFP-T_TgTUBA1. The distinction is important, since this construct was used for SIM. “RFP” with no modifiers would normally indicate the mRFP originally introduced by the Tsien lab, with which SIM imaging of cortical microtubules as shown here would probably be impossible.

We have corrected the text to appropriately specify the fluorescent tubulin.

Reviewer #2 (Remarks to the Author):

The paper entitled ‘Alveolar proteins stabilize cortical microtubules in *Toxoplasma gondii*’ by Harding et al describes the function of a family of multipass transmembrane proteins termed ‘GAPM’ play in the *Toxoplasma gondii*. The authors follow on from previous work demonstrating that these proteins are integral membrane proteins of the IMC and likely hold a particular orientation and most likely function in co-ordinating the stability and position of microtubules (MTs). To prove this the authors use elegant high resolution imaging and molecular genetic techniques and develop a novel method to probe membrane topology.

The work is of very high quality and well interpreted and furthermore, does a good job of dissecting out the function of the GAPM’s away from what could be pleiotropic effects. Further, it applies a new technique to probe the membrane topology of proteins using a chromobody system which will, no-doubt, be used in the field (is this the first time such a technique has been used? If so, important to highlight this).

The only major criticism that I have is that all conclusions on localisation have been drawn using genetically-tagged proteins (ie mCherry or GFP). This is only a real concern as it is clear that the parasites do not like having the C-terminus of GAPM tampered with, as it confers a fitness cost, suggesting the protein function is somehow compromised. For example, genetic tagging could interfere with correct protein trafficking, thus any localisation studies should be interpreted with caution. For example, Figure 1D and E look at substructures of GAPM within parasites, which could quite possibly be artefacts due to protein tagging. I suggest that reporting this level of detail does not provide much benefit due to this possible caveat.

We agree with the reviewer that there are caveats of using fluorescently tagged proteins for localization studies and we have therefore included mention to those caveats in the text, mentioning in the Results that, “*we cannot exclude the possibility that these structures are caused by tagging with fluorescent proteins*” and in the discussion that, “*we cannot rule these structures out as artefacts of fluorescent tagging.*” We nonetheless believe that it is important to report these structures, since markers for the micropore are not currently available, and the localization of GAPMs may provide a clue to these elusive structures.

MINOR POINTS

Figure 1: What basis is the membrane topology predicted? It is stated that this is a prediction in the figure legend but is not discussed at all. This is important to discuss the rationale for this if the topology is predicted to be different between clades.

Topology was reported based on the published analysis [Bullen et al 2009]. This information has been added to the figure legend.

Figure 2: Presents functional data on GAPM1a and TEM images in panel ‘H’ are used to demonstrate defects in membrane morphology upon loss of this gene. I’m unsure of the usefulness of these images as by 18hours the parasites look really unwell and thus effects could be just pleiotropic. The zoom-in box only highlights one area of many that I can see where there is problems with membrane integrity.

We agree that there are severe, pleotropic effects by 18 h including many areas where the membranes do not appear to be arranged correctly. For this reason, we do not use this time point further in the paper. However, similar phenotypes have previously been observed upon disruption of integral IMC proteins [Harding 2016; Beck 2010] using slower conditional systems. We therefore chose to include these observations to relate the GAPM phenotypes to what has been previously observed for other IMC proteins.

Figure 4A: What is the utility of looking at localisation of GAPM in relation to MT upon depletion rather than just showing colocalization at steady state?

Due to the abundance of GAPM1a-AID at the IMC, we rarely observe its colocalization with microtubules at steady state, instead observing an even distribution across the IMC. We believe this is possibly due to our inability to resolve individual IMPs by SR-SIM. In order to clarify this in the text, we have added the sentence: “*In rare cases, co-localisation of GAPM1a-AID with microtubules could be seen in non-treated cells (Fig. 4a, first panel), although it was usually evenly distributed along the entire IMC.*”

Figure 6: There is not enough data to support this model of GAPM. There is no functional data to suggest that GAPMs play a role in anchoring the glideosome (as explored here in Figure 3) and there is no information as to how GAPM’s interact with the MT. I would remove this.

We agree, although many of the proposed interactions are based on published work, and not investigated in this work. We have removed the model as suggested.

This work will be an excellent addition to our understanding of microtubule biology in this early branching eukaryote and thus will be of interest to biologists working on fundamental aspects of apicomplexan biology as well as investigators studying the cytoskeletal dynamics and function.

We appreciate the reviewer thoughtful evaluation of our work.

Reviewers' Comments:

Reviewer #1:

Remarks to the Author:

In the revised manuscript the authors have addressed and completely satisfied all of the issues I had pointed out.

It should now be published forthwith.